# Diversity and Molecular Barcoding of Stink Bugs (Hemiptera: Pentatomidae) Associated with Macadamia in South Africa

**DOI:** 10.3390/insects13070601

**Published:** 2022-06-30

**Authors:** Byron Sonnekus, Bernard Slippers, Brett P. Hurley, Elizabeth Joubert, Michael Stiller, Gerda Fourie

**Affiliations:** 1Department of Biochemistry, Genetics and Microbiology, Forestry and Agricultural Biotechnology Institute (FABI), University of Pretoria, Pretoria 0002, South Africa; byron.sonnekus@fabi.up.ac.za (B.S.); bernard.slippers@fabi.up.ac.za (B.S.); 2Department of Zoology and Entomology, Forestry and Agricultural Biotechnology Institute (FABI), University of Pretoria, Pretoria 0002, South Africa; brett.hurley@fabi.up.ac.za; 3Centre for Excellence, Levubu 0929, South Africa; elsje@centreforexcellence.co.za; 4Biosystematics Division, ARC-Plant Health and Protection, Private Bag X134, Queenswood 0121, South Africa; stillerm@arc.agric.za

**Keywords:** pentatomidae, macadamia, DNA barcoding, phylogenetics, species composition

## Abstract

**Simple Summary:**

Stink bugs inflict extensive damage leading to significant yield and economic loss in the South African macadamia industry. There is currently a need for alternative control strategies to replace the reliance on chemical control in South Africa. Accurate identification and knowledge of species composition are important to inform these management practices. In this study, we identified stink bug species associated with macadamia orchards in the three main growing regions of South Africa. This was performed based on morphology and DNA barcoding. A total of 21 stink bug species were found in macadamia orchards, and *Bathycoelia distincta* was the dominant species found. A group of *Boerias* spp. were found to be dominant in KwaZulu-Natal, and this is the first report of these species associated with macadamia. Evidence of cryptic species diversity was also found within *Pseudatelus raptorius* and an unidentified *Boerias* sp. (*Boerias* sp. 1). Species composition fluctuated over three growing seasons and between growing regions, highlighting the need for ongoing monitoring of these important pest species. The DNA barcode database developed in this study will be valuable for future monitoring, identifications and the implementation of informed management strategies.

**Abstract:**

Stink bugs are major pests of macadamia in South Africa. Accurate identification and knowledge of species composition are important to inform management practices. The overall aims of this study were to identify stink bug species from macadamia orchards in South Africa using morphology, and to establish a DNA database based on the *cytochrome c oxidase subunit 1* gene region. A total of 21 stink bug species were found in macadamia orchards in KwaZulu-Natal, Limpopo and Mpumalanga provinces. *Bathycoelia distincta* Distant, 1878, was the dominant species throughout all three growing regions. Two unidentified species of *Boerias* Kirkaldy, 1909, here designated as *Boerias* sp. 1 and *Boerias* sp. 2, were the second and third most abundant species found in KwaZulu-Natal. No species of *Boerias* has previously been reported in association with macadamia. Evidence of a cryptic third species of *Boerias* was also found. Species composition fluctuated over three growing seasons in Limpopo and differed between the three growing regions during the 2019–2020 season, highlighting the need for ongoing monitoring of these important pest species. The DNA barcode database developed in this study will be valuable for future monitoring and identifications, including cryptic or polymorphic stink bug species and different life stages.

## 1. Introduction

Macadamia trees (*Macadamia integrifolia* Maiden and Betche and *Macadamia tetraphylla* L. Johnson) and their hybrids are planted commercially in various countries, including Australia, the USA (Hawaii), Brazil, Colombia, Guatemala, China, Vietnam, Kenya, Malawi, and South Africa [1]. Production in South Africa has increased rapidly since the 1990s, and the country is currently the largest producer of macadamia nuts globally [1]. The three main growing regions in South Africa include parts of KwaZulu-Natal, Mpumalanga, and Limpopo provinces.

Stink bugs (Hemiptera: Pentatomidae) are an economically important group of insects in agriculture [2,3]. The type of damage inflicted via their piercing/sucking feeding strategy differs between the type of crop, the phenological stage of the host plant and the species of stink bug involved. Typical symptoms of stink bug damage on nut crops such as macadamia, almonds (*Prunus dulcis* (Miller) D.A. Webb), hazelnuts (*Corylus* sp.), pecan (*Carya illinoinensis* Koch) and pistachio (*Pistacia vera* Linnaeus) include premature nut drop, malformed nuts, discoloured spots, and nut necrosis [4,5,6,7,8,9].

A complex of stink bug species has been reported to occur in macadamia orchards in South Africa [9,10,11]. They are regarded as the largest contributing factor to macadamia kernel damage in South Africa, causing an annual loss of about USD15.23 million [12]. The previously recorded species include *Bathycoelia distincta* Distant, 1878 (two-spotted stink bug), *Nezara viridula* (Linnaeus, 1758) (green vegetable bug), *Chinavia pallidoconspersa* (Stål, 1858) (yellow-edged stink bug, formerly *Nezara pallidoconspersa*), *Parachinavia prunasis* (Dallas, 1851) (formerly *Acrosternum prunasis* and *Nezara prunasis*), and *Pseudatelus raptorius* (Germar, 1838) (powdery stink bug, formerly *Atelocera raptoria*). *Bathycoelia distincta* is thought to be the most dominant and damaging species [10,11]. Variation in stink bug species composition in South African macadamia orchards over time has been reported, and the number of different species present has increased drastically in recent years [9,11].

Accurate pest identification is crucial for the successful development and implementation of pest management strategies, such as biological and behavioural control [13,14]. DNA barcoding is a valuable tool for insect pest identification to complement morphological identification [15,16,17]. Morphological identification can also be complicated by the lack of easily identifiable features in some of the life stages (i.e., eggs and nymphs), colour morphs, and declining taxonomic expertise. For these reasons, a DNA sequence database for agricultural pests of concern can have much value. Such a database does not currently exist for stink bugs associated with macadamia in South Africa.

The aims of this study were to identify stink bug species present in macadamia orchards from the three main growing regions of South Africa and to develop a DNA barcoding database of these stink bug species. To achieve this, stink bugs were collected over three consecutive seasons from the Limpopo province and for one season from Mpumalanga and KwaZulu-Natal provinces and identified based on morphology. The *cytochrome c oxidase subunit 1* (CO1) gene region of a representative set of specimens was obtained and used to infer phylogenetic relationships of the species and determine genetic divergence within and between the abundant species [18,19]. Additionally, the presence or absence of each species was compared between the regions and across three growing seasons in Limpopo.

## 2. Materials and Methods

### 2.1. Insect Collections

Stink bug specimens were collected over three growing seasons from 2017 to 2020 via farmer scouting, from nut set to harvest. Farmer scouting involves the application of a broad-spectrum insecticide spray (i.e., pyrethroid) to about ten trees per orchard before dawn and subsequent collection of insects on plastic sheets on the ground within three to four hours after spraying. In both the 2017–2018 and 2018–2019 seasons, samples were collected from two farms in Levubu, Limpopo. For the 2019–2020 season, stink bug specimens were collected from two different farms in the Limpopo, KwaZulu-Natal and Mpumalanga provinces (Figure 1). The same farms were sampled in Limpopo across all seasons. Stink bug specimens were preserved in 70% ethanol at −20 °C for subsequent analyses.

### 2.2. Morphological Identification

Scout batches from KwaZulu-Natal, Limpopo and Mpumalanga were sorted by date and location. A scout batch included all the specimens obtained from all ten trees collected on a specific date. Stink bug specimens were assigned to putative species groups based on external morphological features described in taxonomic keys available in the literature [20,21,22,23,24,25]. The other species, for which keys are not readily available, were identified based on comparisons of morphological characteristics to verified species in the National Collection of Insects, Pretoria, Agricultural Research Council Plant Health and Protection. Examples of these morphological characteristics include body size, body colour, antennae, stylets, connexivum, wing membrane, scutellum, pronotum, anterior and posterior margins, and legs. Voucher specimens were pinned, accession numbers assigned and deposited in the National Collection of Insects, Pretoria, Agricultural Research Council-Plant Health and Protection (PENT00006–PENT00069). A representative from each morphological group was photographed.

The number of specimens obtained per species was calculated (Appendix A). The presence of each species per scout batch from each location was also determined (Appendix A). Species were subsequently grouped as common or rare, as well as being present throughout the season, early (prior to the nut developing a thick husk and shell) or late (thick husk and shell which develops as the nuts mature) during nut phenological development. Species were regarded as common if the species were present in all three growing regions or in more than 20% of the scout batches obtained in a growing region.

### 2.3. DNA Extraction, Amplification and Sequencing

Genomic DNA was extracted from the legs and pronotum area of each representative stink bug specimen using the NucleoSpin^®^ DNA insect kit (Separations, Johannesburg, South Africa). The DNA concentration was quantified using a Nanodrop spectrophotometer. PCRs were performed in a 25 µL volume containing 5 µL of 5 mM MyTaq™ reaction buffer containing MgCl_2_ and dNTPs (BioLine, South Africa), 0.5 µL of MyTaq™ DNA polymerase, 0.5 µL of 10 mM of LCO 1490 and HCO-2198 [26], and 50–100 ng of DNA. Thermocycling conditions included an initial denaturation at 94 °C for 1 min, followed by 5 cycles of 94 °C for 1 min, 45 °C for 1 min 30 s and 72 °C for 1 min 30 s. This was followed by 30 cycles of 94 °C for 1 min, 50 °C for 1 min 30 s and 72 °C for 1 min, with a final hold of 5 min at 72 °C. The amplicons obtained were subjected to agarose gel electrophoresis on a 2% agarose gel in a 1× sodium borate buffer and visualized using GelRed (Biotium, Hayward, CA, USA) and a 100 bp DNA ladder (Promega^®^, Madison, WI, U.S.A) to determine the size of each amplicon. A Molecular Imager^®^ GelDoc™ (Bio-Rad, Hercules, CA, USA) was used to visualize the amplicons under ultraviolet light. Following the confirmation of the presence of the correct amplicons, a PCR clean up reaction was performed using ExoSAP-IT™ (Applied Biosystems, Foster City, CA, USA) by following the manufacturer’s protocol.

The PCR products were sequenced in both directions using the BigDye Terminator v3.1 Cycle Sequencing Kit (Applied Biosystems, Port Elizabeth, South Africa). The thermal cycler was programmed such that initial denaturation was at 96 °C for 2 min, followed by 25 cycles at 96 °C for 10 s, 50 °C for 5 s and 60 °C for 4 min. The samples were precipitated using sodium acetate and sequenced using an ABI 3100 Automated Capillary DNA Sequencer (Applied Biosystems, Waltham, MA, USA) at the Bioinformatics Sequencing facility at the University of Pretoria.

### 2.4. Phylogenetic Analyses

Forward and reverse sequence reads were checked for base calling accuracy and manually edited using Biological Sequence Alignment Editor (BioEdit) [27]. Consensus sequences for each individual were generated by aligning the forward and reverse sequences. Sequences were translated into amino acids to screen for the presence of premature stop codons. The generated sequences were then subjected to Basic Local Alignment Search Tool (BLAST) searches against the National Center for Biotechnology Information (NCBI) (www.ncbi.nlm.nih.gov) (accessed on 12 January 2021) and Barcode of Life Data Systems (BOLD) (www.boldsystems.org) (accessed on 12 January 2021) databases. Representative stink bug species sequences were downloaded from BOLD and NCBI, representing 13 genera within the subfamily Pentatominae [28,29,30,31,32,33,34,35,36,37,38,39]. Thereafter, sequences were aligned using Multiple Alignment using Fast Fourier Transform (MAFFT) version 7 (https://mafft.cbrc.jp/alignment/software) (accessed on 13 January 2021) [40] and trimmed to a length of 472 bp. The aligned sequences were subjected to phylogenetic analysis based on maximum likelihood with 1000 bootstrap replicates. Phylogenetic analysis was performed using Randomized Axelerated Maximum Likelihood (RaxML) version 8 [41] and viewed/edited in Mega 7.0 [42]. *Graphosoma lineatum* (Linnaeus, 1758) (Pentatomidae: Podopinae) (KM022369) was used as an outgroup. Sequences obtained in this study were added to the Genbank database (accession numbers OM416463–OM416523).

### 2.5. Sequence Divergence Analysis

Sequence divergence was calculated for all the common stink bug species. Interspecific divergence was used to determine whether the stink bug specimen groupings, based on morphology, were correct. Intraspecific divergence was determined to assess the possible presence of cryptic species. Nucleotide diversity (π), average number of nucleotide substitutions per site (D_xy_), and net nucleotide substitutions per site (D_a_) were calculated using DNA Sequence Polymorphism (DnaSP) version 5.10.01 [43]. The nucleotide diversity (π) value is the mean number of nucleotide differences per site between two sequences [44]. For comparative purposes π, D_xy_ and D_a_ values were converted to percentages.

## 3. Results

### 3.1. Morphological Identification

A total of 5520 stink bug specimens were collected from 342 scout batches from farms in the three main growing regions of South Africa (Appendix A). This included 166 batches from KwaZulu-Natal and 144 from Limpopo, collected over three seasons, and 32 from Mpumalanga, collected over one season. The specimens were subsequently grouped, based on morphology, into 21 stink bug species, representing two Pentatomidae subfamilies (Pentatominae and Asopinae) (Figure 2 and Figure 3, Appendix A). Of these species, *Boerias* spp. had two distinct morphological groups but could not be identified as species. The first group consisted of individuals that possessed an orange to brown anterolateral margin and a white/brown granulated abdominal venter. The second group had individuals with a variegated connexivum and a smooth white-to-cream abdominal venter.

### 3.2. Species Composition

#### 3.2.1. Species Composition across Growing Regions

During the 2019–2020 growing season, a total of 18 species were found in KwaZulu-Natal, while a total of 8 species and 10 species were present in Limpopo and Mpumalanga, respectively (Figure 4). Based on the total number of specimens collected, *B. distincta* was the dominant species found in Limpopo (88%), KwaZulu-Natal (76%) and Mpumalanga (62%) (Appendix A). In terms of species presence per scout batch, *B. distincta* also represented the most prevalent species in Limpopo, and Mpumalanga, followed by *N. viridula* (Appendix A). However, in KwaZulu-Natal, species of *Boerias* were more prevalent than *B. distincta*.

*Bathycoelia distincta*, *C. pallidoconspersa*, *N. viridula*, *P. prunasis*, *Parantestia* sp. (Linnavuori, 1973), and *P. raptorius* were commonly found in scout batches from all three growing regions. *Boerias* sp. 1 and *Boerias* sp. 2 were found in scout batches of KwaZulu-Natal, while *Piezodorus* sp. (Fieber, 1860) was common in scout batches from Limpopo and Mpumalanga. Scout batches also included nymphs of *B. distincta* and *P. raptorius* that represent the two Pentatomidae species that have been reported to reproduce in macadamia orchards in South Africa to date [9]. *Bathycoelia distincta*, *Boerias* sp. 1, *Boerias* sp. 2, *C. pallidoconspersa*, *N. viridula*, *P. prunasis*, *Parantestia* sp., *Piezodorus* sp. and *P. raptorius* were present throughout the season.

An additional twelve species, only observed in less than 20% of scout batches per region, were also detected in this study (Appendix A). These included *Agonoscelis versicoloratus* (Turton, 1802), *Anolcus campestris* Bergroth, 1893, *Antestia* sp., *Aspavia albidomaculata* (Stål, 1853), *Antestiopsis thunbergii* (Gmelin, 1790), *Basicryptus costalis* (Germar, 1838), *Carbula recurva* Distant, 1915, *Caura rufiventris* (Germar, 1838), *Coenomorpha nervosa* Dallas, 1851, *Macrorhaphis acuta* Dallas, 1851, *Platacantha lutea* (Westwood, 1837) and *Tripanda signitenens* (Distant, 1898). *Anolcus campestris*, *A. thunbergii* and *M. acuta* were present early in the season in KwaZulu-Natal, while *P. lutea* was found in Mpumalanga and KwaZulu-Natal in the early season. *Caura rufiventris* was present in the late season and was unique to Limpopo. *Antestia* sp. and *C. nervosa* were present throughout the season in Mpumalanga and KwaZulu-Natal. *Agonoscelis versicoloratus*, *A. albidomaculata*, *B. costalis*, *C. recurva* and *T. signitenens* were also present throughout the season and were unique to KwaZulu-Natal during the 2019–2020 season. *Agonoscelis versicoloratus*, *C. recurva* and *T. signitenens* were, however, also found in previous seasons in Limpopo in this study (Figure 5).

#### 3.2.2. Species Composition across Growing Seasons in Limpopo

There was a clear distinction in species composition across three growing seasons in Limpopo. In total, 15 stink bug species were present across three seasons, with 11 of them present in 2017–2018, 14 in 2018–2019 and 8 in 2019–2020 (Figure 5). In terms of species presence per scout batch, *B. distincta* was the most prevalent species found in scout batches obtained from Limpopo across two seasons, namely 2018–2019 and 2019–2020. *Nezara viridula* was the species observed the most in the scouting batches during 2017–2018 (Appendix A). *Parantestia* sp. and *N. viridula* were the second most prevalent species, according to the number of scout batches where they were found, for 2018–2019 and 2019–2020, respectively.

### 3.3. Phylogenetic Analysis

A 650 bp fragment of the mtDNA COI gene was sequenced for 61 stink bug specimens collected during this study (Appendix A). These specimens included representatives from each morphological group. Based on percentage identity values of 98% or higher, BLAST searches against the NCBI database identified some of the individuals as *N. viridula* (n = 6) and *Piezodorus* sp. (n = 4). A search against the Barcode of Life Database (BOLD) systems identified the same two species, as well as *Agonoscelis puberula* (Stål, 1853) (n = 2) and *M. acuta* (n = 2). Thus, sequences for the remaining 18 stink bug species were the first to be submitted to Genbank for these species.

Maximum likelihood analysis resolved the sequences into two distinct subfamily clades, namely Pentatominae and Asopinae (Figure 6). Within the Pentatominae clade, all 31 of the different species were separated into their individual clades, with strong bootstrap values (>90%), and represented 24 genera. TheAsopinae species included in the analysis, namely *M. acuta*, grouped within the Asopinae subfamily clade.

### 3.4. Sequence Divergence Analysis

Individuals of each species clustered within their respective clade. The π values for *B. distincta*, *C. pallidoconspersa*, *N. viridula*, *P. prunasis*, *Parantestia* sp. and *Piezodorus* sp. were 0.43, 0.11, 0.26, 0.11; 0.26 and 0.86%, respectively (Table 1). The π values for the *Boerias* sp. 1 and *P*. *raptorius* groups were found to be high, with approximate values of 4.71% and 4.28%, respectively, which suggest the presence of cryptic species. This was congruent with the phylogenetic grouping of *Boerias* sp. 1, which indicates the presence of two species within this clade (Figure 6). The genetic divergence between species groups, i.e., the average number of nucleotide substitutions per site (D_xy_) and the net nucleotide substitutions per site (D_a_) between populations [44] for the same species groups ranged from 11.4–17.7% (0.114–0.177) and 8.7–15.4% (0.087–0.154), respectively.

## 4. Discussion

Based on morphological identification, 21 stink bug species were found in macadamia orchards in KwaZulu-Natal, Limpopo and Mpumalanga. This study serves as the most extensive species list to date, and it is the first survey conducted in KwaZulu-Natal orchards. *Bathycoelia distincta* continues to be the most dominant stink bug species found in macadamia orchards in South Africa. The survey also led to the discovery of *Boerias* spp. occurring in a high proportion of scout batches in KwaZulu-Natal. Nine species were commonly found in macadamia orchards throughout the season, with 12 species regarded as rare. Species composition differed between growing regions and between different growing seasons. For all Pentatomidae species found, a COI DNA barcode was established as only four COI sequences were available for the 21 stink bug species detected in this study. Sequence divergence calculations were used to determine the accuracy of morphological groupings and identifications. All species groups represent distinct species except for *Boerias* sp. 1 and *P. raptorius,* which appear to contain cryptic species.

A high diversity of stink bug species was found across the main growing regions of South Africa. Nine stink bug species are reported from macadamia orchards in South Africa for the first time, namely *A. campestris*, *A. thunbergii*, *Boerias* sp. 1, *Boerias* sp. 2, *C. recurva*, *C. rufiventris*, *Parantestia* sp., *P. lutea* and *T. signitenens.* In addition to species diversity, differences in species composition across seasons and growing regions were also recorded in this study. The high diversity of species and fluctuation of species presence across and between growing seasons are in agreement with previous studies that investigated stink bug species in macadamia orchards in South Africa [9,11,45]. Heteropteran populations are known to be influenced by factors such as tree size, tree and canopy density, surrounding natural vegetation, and differences in husbandry practices [46]. Changes in species diversity and composition could also be due to differences in host phenology at the time of sampling and differences in temperature and climate between growing regions and seasons. Therefore, a more in-depth study including more sampling sites, climatic conditions of the sampling sites and thermal biology of the stink bug species is needed.

*Bathycoelia distincta*, *Boerias* sp. 1, *Boerias* sp. 2, *C. pallidoconspersa*, *N. viridula*, *P. prunasis*, *Parantestia* sp., *Piezodorus* sp. and *P. raptorius* were identified as common species present throughout the season. This suggests that macadamia is a suitable food source for these stink bug species, although the damage potential of *Boerias* spp., *P. prunasis*, *Parantestia* sp. and *Piezodorus* sp. is currently unknown. In addition to macadamia, *N. viridula* has been reported as a pest of hazelnut and pecan in Europe and the United States [5,47]. *Pseudatelus raptorius* is also considered a major pest of pistachio in the Northern Cape Province of South Africa [48] and a pest of pecan, litchis and avocados in the Mpumalanga province [49,50].

Species of *Boerias* were the most prevalent species in scout batches obtained from the KwaZulu-Natal province. Little is currently known regarding the biology of the species and their capability to inflict nut damage. The stylets of the *Boerias* spp. are shorter than those of *B. distincta* (Figure 2). The hypothesis has been that stink bug species with longer stylets are able to penetrate deeper and inflict damage on more well-developed fruits/nuts and are thus able to inflict damage throughout the season [45,51]. Stylet penetration potential, however, has been shown to be dependent on specific rostral segments and not total rostral length, and species with shorter stylets may, therefore, also cause late damage [52,53]. For example, *C. pallidoconspersa* and *Antestia* sp. have been reported to be capable of inflicting late damage on macadamia nuts despite their relatively shorter mouthparts [54,55].

A total of 12 species were considered rare since they were found in low numbers in very few of the scout batches obtained from the different growing regions. The pest status for only a few of these species is currently known. *Antestia* sp. and *A. versicoloratus* feed on flowers and fruits of *Jatropha curcas* L. (physic nut) in West Niger [56]. *Antestiopsis thunbergii* is a major pest of coffee plants in Africa [57]. *Coenomorpha nervosa* is a pest of avocado in Mpumalanga and a major pest of pecan in the Northern Cape and North West provinces in South Africa [58]. *Coenomorpha nervosa*, however, was recently shown to not inflict significant macadamia nut damage [54]. *Macrorhaphis acuta* is classified within the predatory subfamily Asopinae and is a predator of *Achaea lienardi* (Boisduval, 1833) (Lepidoptera: Noctuidae) larvae in South Africa [59].

In addition to species identification based on morphology, COI sequences were obtained to supplement identifications of morphological groupings. This allowed for the establishment of a COI sequence database for the stink bug species found in macadamia orchards in South Africa. A total of 61 COI sequences were produced for the 21 species, of which 18 previously lacked sequence data. This database will be used as a diagnostic tool for future identification of stink bug species and their relevant life stages. This is of particular importance since morphological identification of insects can be difficult and expert knowledge of the use of taxonomic keys is often required [15,16]. The different life stages for several of the stink bug species identified in this study are currently unknown, and a COI barcode can help link an adult to a nymphal stage. COI sequences were obtained for nymphs of *B. distincta* and *P. raptorius,* and the nymphs grouped within their respective clades. Polymorphism is also known to occur in various stink bug species such as *N. viridula* [60]. Two different colour morphs were observed for individuals of *N. viridula* collected during this study. These specimens were successfully resolved into the same clade, despite their morphological difference, by using COI DNA barcoding and a phylogenetic approach.

The genetic divergence of the respective species analyzed in this study ranged from 0 to 0.86%, except for the *Boerias* sp. 1 and *P. raptorius* groups, of which both had values over 4%. Genetic divergence of ≥2% among individuals of an insect group may indicate the presence of cryptic species [18,61]. The high intraspecific divergence value between *Boerias* sp. 1 and *Boerias* sp. 2 was substantiated by the distinct phylogenetic groups obtained within the group (Figure 6). Morphological differences for individuals within the *P. raptorius* and *Boerias* sp. 1 were not distinct and required further work to formally describe the cryptic species within these groups. Mean COI divergence between congeneric insect species pairs ranges from 6.6 to 11.5% (Hebert et al., 2003b).

## 5. Conclusions

In summary, a high diversity of stink bug species, with fluctuation of stink bug species presence, both over time and across growing regions, was observed in macadamia orchards in KwaZulu-Natal, Limpopo and Mpumalanga. This highlights the importance of continuous monitoring of stink bug presence and assessment of their damage potential. The DNA barcoding database established in this study will assist in accurate pest identifications in the future. This is of utmost importance for effective pest management, especially when implementing species-specific control methods. This could also apply to other cropping systems, not just macadamia, in Southern Africa, as stink bugs are known to be pests of various crops in this region. The novel association of *Boerias* spp. in a high proportion of scouting samples from KwaZulu-Natal throughout the season indicates that this insect group plays a potential role in macadamia nut damage and thus requires further investigation.

## Figures and Tables

**Figure 1 insects-13-00601-f001:**
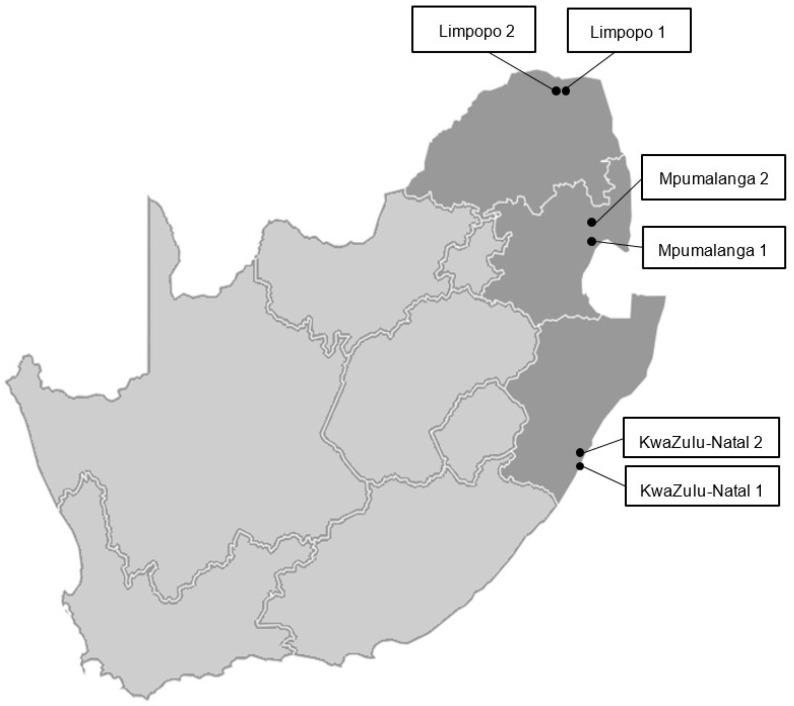
Map of South Africa showing the farm locations where stink bugs were sampled in KwaZulu-Natal, Limpopo and Mpumalanga (indicated in grey).

**Figure 2 insects-13-00601-f002:**
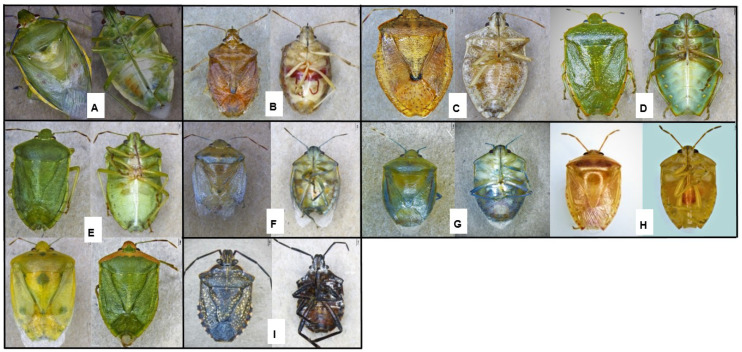
Stink bug species found in high proportions during the 2019–2020 season in all three growing regions. (**A**). *Bathycoelia distincta* (**B**). *Boerias* sp. *1* (**C**). *Boerias* sp. *2* (**D**). *Chinavia pallidoconspersa* (**E**). *Nezara viridula* (**F**). *Parachinavia prunasis*. (**G**). *Parantestia* sp. (**H**). *Piezodorus* sp. (**I**). *Pseudatelus raptorius*.

**Figure 3 insects-13-00601-f003:**
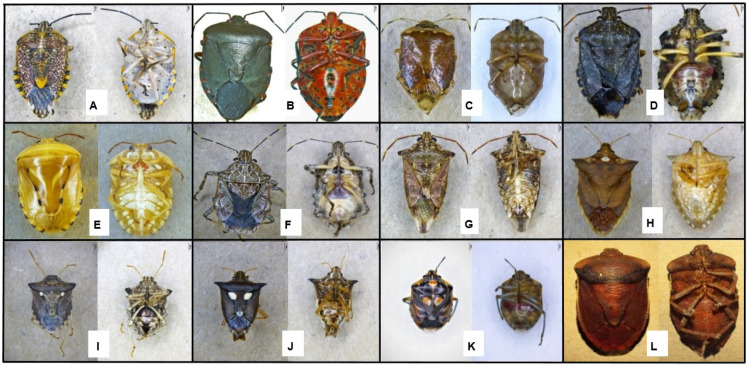
Stink bug species found in low proportions during the 2019–2020 season in all three growing regions. (**A**). *Agonoscelis versicoloratus* (**B**). *Caura rufiventris* (**C**). *Antestia* sp. (**D**). *Coenomorpha nervosa* (**E**). *Platacantha lutea* (**F**). *Anolcus campestris*. (**G**). *Macrorhaphis acuta* (**H**). *Tripanda signitenens* (**I**). *Carbula recurva* (**J**). *Aspavia albidomaculata* (**K**). *Antestiopsis thunbergii* (**L**). *Basicryptus costalis*.

**Figure 4 insects-13-00601-f004:**
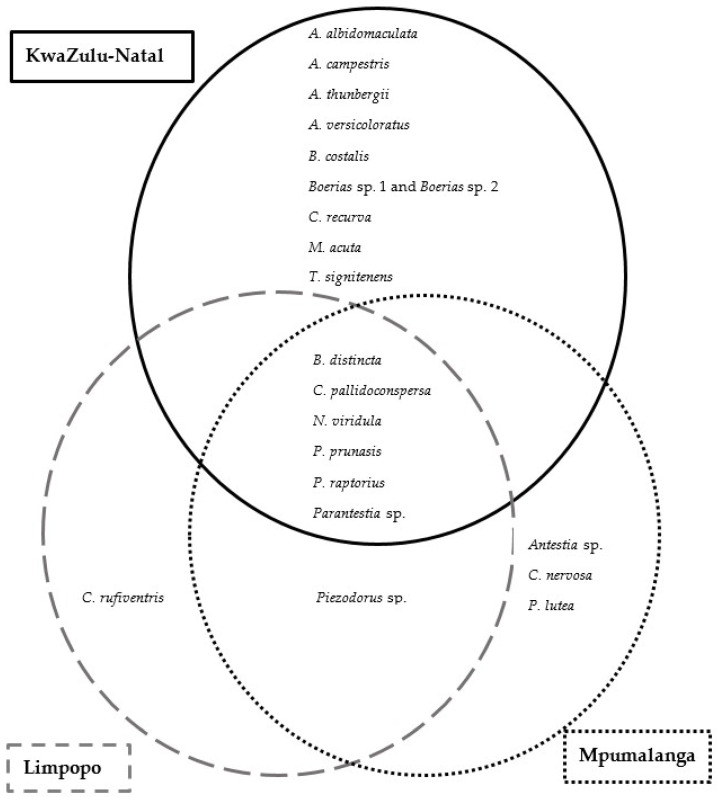
Stink bug species presence in KwaZulu-Natal, Limpopo and Mpumalanga for the 2019–2020 growing season.

**Figure 5 insects-13-00601-f005:**
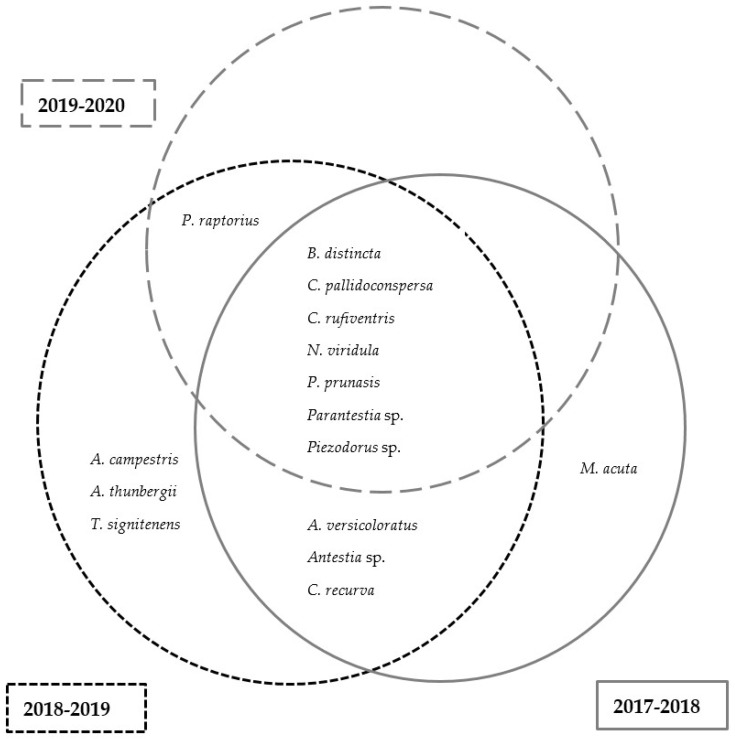
Stink bug species presence across three growing seasons (2017–2020) in the Limpopo growing region.

**Figure 6 insects-13-00601-f006:**
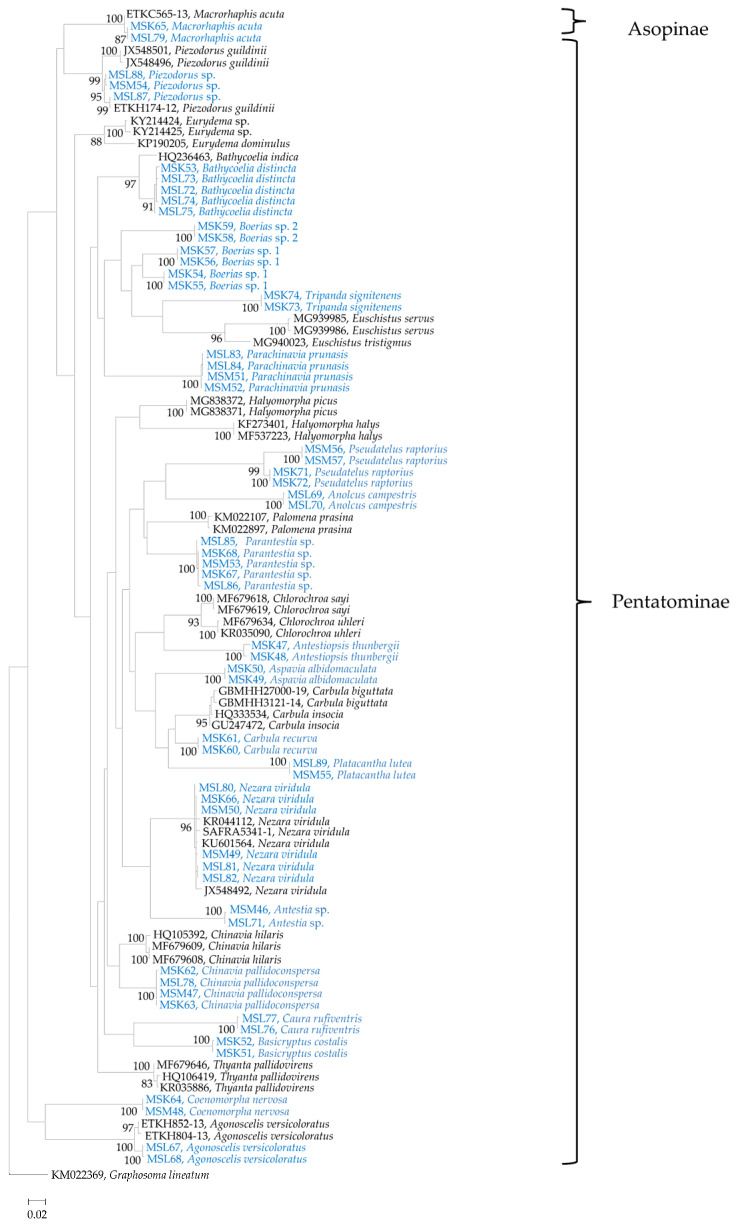
Maximum likelihood stink bug phylogeny based on the COI gene region. The specimens in blue represent species of stink bugs obtained from macadamia orchards in the KwaZulu-Natal, Limpopo and Mpumalanga provinces of South Africa in this study.

**Table 1 insects-13-00601-t001:** Nucleotide diversity (π) values for nine of the stink bug species from this study.

Stink Bug Species	Nucleotide Diversity (π)	%
*Bathycoelia distincta*	0.00428	0.43
*Boerias* sp. 1	0.04711	4.71
*Boerias* sp. 2	0	0
*Chinavia pallidoconspersa*	0.00107	0.11
*Nezara viridula*	0.00257	0.26
*Parachinavia prunasis*	0.00107	0.11
*Parantestia* sp.	0.00257	0.26
*Piezodorus* sp.	0.00857	0.86
*Pseudatelus raptorius*	0.04283	4.28

## Data Availability

Sequence data openly available in Genbank (www.ncbi.nlm.nih.gov) (accessed on 27 January 2022).

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
