# Peer review of "Diversity and Molecular Barcoding of Stink Bugs (Hemiptera: Pentatomidae) Associated with Macadamia in South Africa"

_insects, 2022, doi:10.3390/insects13070601_

Round 1

Reviewer 1 Report

The authors investigated the species diversity of stink bugs associated with macadamia in South Africa and assessed the fluctuation of species composition over growing seasons and between growing regions. Moreover, they established a DNA barcode database which would be valuable for future monitoring, identifications and management of stink bugs. In general, the manuscript is of important significance and well-written. I have some minor comments.

Comment 1. Lines 278-279: “All the nine common species were present throughout the season”. It’s unclear what the authors mean. Does the “nine common species” refer to “Bathycoelia distincta, C. pallidoconspersa, N. viridula, P. prunasis, Parantestia sp., P. raptorius, Boerias sp. 1, Boerias sp. 2 and Piezodorus sp.”? If yes, please clearly state this.

Comment 2. Table 3: Table 3 requires a bit more explanation. Please explain the meaning of number 8 on the table. Which species does the number 8 refer to?

Comment 3. Lines 207-208: “Podisus maculiventris (Say, 1832) (Pentatomidae: Asopinae) (MG940398) was used as an outgroup.” According to the results of morphological identification and DNA barcoding, the specimens were grouped to Pentatominae and Asopinae. Why did the authors use the species belonging to Asopinae as outgroup?

Comment 4. Lines 462-464: “The high intraspecific divergence value for Boerias sp. 1 was substantiated by the distinct phylogenetic groups obtained within the group (Figure 6)”. Actually, MSK54 and MSK55 clustered with MSK56 and MSK57. In this study, the samples comprised stink bug species from different genus and subfamily. I think it would be more convincing using more molecular genes in the maximum likelihood analysis. For example, the combined dataset of mitochondrial COI, COII and Cytb markers, and nuclear marker EF1α is frequently employed in the phylogenetic analysis. In addition, it’s unclear which branch the bootstrap values represent. Please modify the position of bootstrap values on the Figure 6.

Author Response

Response to Reviewers’ Comments and Suggestions

Please note that the page and line numbers referred to in our responses represent those found in the newly submitted manuscript. Furthermore, specific changes are indicated in italics, where appropriate.

Reviewer 1:

  1. The reviewer requested the authors to be clear when referring to the nine common species referred to in lines 276-278.
  • This is a good suggestion by the reviewer, and we have included the species names for the nine common species
  • L276-278: “All the nine common species were present throughout the season.” was changed to “Bathycoelia distincta, Boerias sp. 1, Boerias sp. 2, C. pallidoconspersa, N. viridula, P. prunasis, Parantestia sp., Piezodorus sp. and P. raptorius were present throughout the season.”
  1. The reviewer indicates that species number 8 was missing in the key to Table 3.
  • We are grateful to the reviewer for picking up this error, but Table 3 has been removed as suggested by Reviewer 2.
  1. The reviewer asked why a species belonging to Asopinae was used as an outgroup.
  • Podisus maculiventris was originally used as an outgroup, the same outgroup used in Barman et al. 2017, before Macrorhaphis acuta was added to the phylogeny. We agree with the reviewer that a different species should be used as an outgroup. Graphosoma lineatum (KM022369), from the subfamily Podopinae, was used as an outgroup instead.
  • Lines 205-206: “Podisus maculiventris (Say, 1832) (Pentatomidae: Asopinae) (MG940398)” was changed to “Graphosoma lineatum (Linnaeus, 1758) (Pentatomidae: Podopinae) (KM022369)”. Please also see new figure 6.
  1. The reviewer indicated that the statement made in lines 451-453 was incorrect due to the clustering of MSK54 and MSK55 with MSK56 and MSK57 in Figure 6. The reviewer suggested the use of more molecular genes in the maximum likelihood analysis. The reviewer requested us to modify the bootstrap values on Figure 6.
  • The error of the statement referring to intraspecific divergence was corrected by referring to the intraspecific values between the two Boerias instead of within the Boerias sp. 1 group.
  • Lines 451-452: “for Boerias 1” was changed to “between Boerias sp. 1 and Boerias sp. 2
  • We agree with the reviewer that more gene regions should be used for the maximum likelihood analysis. However, the purpose of this study was to establish a COI database as a starting point since there were no sequences previously available for most of the stink bug species in this study. It would be valuable to include more gene regions in future studies to improve the accuracy of the maximum likelihood analysis.
  • The position of the bootstrap values on Figure 6 were modified to make it clear which branch the bootstrap values represent.

Reviewer 2 Report

The authors have obtained standardized samples of macadamia-associated insects at a total of 6 sites in 3 South African provinces across 3 growing seasons, and have attended to the pentatomids.  They report morphological ID's and CO1 sequences, and give an account of the relative abundance and distribution of the species.

This study is valuable for its standard sampling method, allowing the authors to characterize the relative abundance of the species and their distribution.  The fauna was poorly documented with respect to  CO1 sequences prior to this study.  On the whole this is a well justified, thorough, and well-presented investigation.

I have a few substantive concerns and a number of minor copy-editing suggestions.

1. My main concern is with voucher specimens.  Ideally, every specimen from which a sequence was obtained should be permanently retained as a museum voucher.  That way, there is a permanently well-documented link between the sequence and the specimen's morphology.  The authors report sequences for 61 individuals but voucher codes for only 33, and it's impossible to tell which 33 these were.  The authors need to add a column to Table 1 for the voucher codes cited on lines 153-154, so that the reader can see which specimens have vouchers.  Ideally, they should also submit the other 28 sequenced specimens as vouchers as well.

2. Methods.  Section 2.2 and a half. How were individuals chosen for sequencing?  For instance: "One individual of each species from each site where the species occurred was selected for sequencing.  If the species was present at only a single site, 2 specimens were used." That would have made some sense, but that's not exactly what the authors did.  I can't quite reconstruct the sampling scheme and selection criteria.  Please describe.

3. I'm confused about the number of different DNA sequences obtained from each species.  Table 2 gives non-zero values for the nucleotide diversity of most of the species, and yet each of these is represented by a single DNA sequence in the phylogenetic analysis (Fig. 6).  What's going on?  Were sequences omitted from the phylogenetic analysis? Why doesn't Fig. 6 show clusters of closely related species for Bathycoelia distincta and the other species with non-zero pi values in Table 2?

4. The authors should consider adding another table, possibly as supplemental information, that would give precise counts of all morphospecies for each sample, or at least each site and season.  That would leverage the power of the standardized sampling design and allow subsequent quantitative analyses of abundance and diversity.

5. The first few references to cryptic and incompletely identified species were very confusing.  Having read the whole paper, I think I can make sense of most of them and can offer editing suggestions.  Specifically:

5a.  Lines 22-23, "A cryptic species of Boerias sp. 1 and Pseudatelus raptorius were also identified."  There are a lot of confusing things about this sentence.  One is the apparent mismatch in number (A cryptic species. . . were identified.) Another is characterizing as "identified" an entity called "a cryptic species of Boerias sp. 1"; on 234 it is reported more accurately that some species -- presumably including this one -- could not be identified. Change this sentence to something like:  "Evidence of cryptic species diversity was also found, within Pseudatelus raptorius and an unidentified species of Boerias." Or "Evidence of two additional, cryptic species was also found, one within Pseudatelus raptorius and another within an unidentified species of Boerias (Boerias sp. 1)."

5b. Lines 34-36, "Boerias sp. 1 Kirkaldy, 1909 and Boerias sp. 2 Kirkaldy, 1909 were the second most abundant species found in KwaZulu-Natal and this is the first report of these species associated with macadamia."  There is more weirdness here with respect to number.  How can two species be the second most abundant species?  The second and third most abundant species?  It's almost as if you haven't actually sorted them out into these 2 separate morphospecies before counting them. (If that's what's going on, please do count them separately or explain why you can't.)  Also, I'm pretty sure Kirkaldy did not describe either Boerias sp. 1 or Boerias sp. 2 in 1909.  Omit the author name and date.  If you feel you must include the author of the genus, write something like "Two unidentified species of Boerias Kirkaldy, 1909, here designated as Boerias sp. 1 and Boerias sp. 2." Also, it's kind of weird to claim that this is the first report of these species associated with macadamia, when you haven't identified the species.  You aren't really reporting the presence of any particular species on macadamia here, as you are not purporting to be able to identify the species.  Maybe change this to something like "No species  of Boerias has previously been reported in association with macadamia" or something, if that is what is meant.   [By the way, I am taking the authors literally when they say these could not be identified to species.  Do you simply  not have the resources to ID these to species?  Or do you think that they represent undescribed species?  If the latter, please characterize them as undescribed, rather than unidentified.]

5c. "A third cryptic Boerias sp. was also identified on macadamia in this region."  This is another weird sentence.  So far in the abstract you haven't mentioned any cryptic species of Boerias.  So how is this the third one?  But it is a third species of Boerias, so this aspect may be reparable with a simple comma: "A third, cryptic Boerias sp. was also identified on macadamia in this region."  That version makes it clear that you are talking about a third species of Boerias, that, by the way, happens to be cryptic.  But it's still weird to say you have "identified" something that remains avowedly unidentified.  How about something like: "Evidence of a cryptic third species of Boerias was also found."

6. Table 3.  Please consider omitting this table.  It adds little.  And as currently phrased, the caption is wildly misleading.  It implies that it shows the divergence between populations within species, but it shows nothing of the kind.  All the divergences are interspecific.  No doubt my strong dislike of this table is due to my  having been deceived and confused by the current caption.  But even with a repaired caption the table would seem to have little interest or utility.

Minor suggestions and copy-edits:

line 33.  "Bathycoelia distincta Distant, 1878 was the dominant species throughout all three growing regions."  Consider putting a comma after 1878.  I realize that many entomology journals call for a comma between an author and year as part of their house style, and that many authors and editors routinely publish sentences like this one, with just the single comma.  But the house style of a journal should not trump the rules of standard punctuation in English, and in actual English you really can't put a single comma in the middle of a subject or any other unitary phrase.  Putting a second comma after 1878 solves the problem.

line 49,  change "Hawaii"  to "USA (Hawaii)" or otherwise re-phrase.  [Hawaii is not a country.]

lines 58-59, remove the comma between the specific epithet and the author name, for each species.  Also remove the genus author name (Linnaeus) after "Corylus sp."

line 64, change "previous" to "previously"

lines 68-69, "The species identity of some observed species is however unknown."  You really do need to set off "however" with commas here.  That was what I initially tripped over when reading it.  But do you need this sentence at all?  After all, here you are not reporting results, you are reviewing the literature.  Do you need to say that not everyone who sees stinkbugs can ID them?  The sentences is pretty confusing in context.  Omit?

line 74, omit "approaches"

lines 78-79, change "DNA a sequence database" to "a DNA sequence database"

lines 88, change "the species presence" to "the presence or absence of each species"

line 151, change "in literature" to "in the literature"

line 234, change "to species-level" to "to species"

line 339, "Individuals of each species group clustered well within their respective branches."  This is a difficult sentence to interpret.  Does "well" modify "clustered" or "within"?  And what does "within their respective branches" mean, anyway?  Also, what is a "species group"?  I will refrain from suggesting an alternative wording, due to my confusion expressed above in point #3.  You say the sequences for each species cluster, but I see no clusters in Fig 6.  I see only a single sequence for each species.

line 421-422, delete "which requires further investigation". [This statement may be redundant, given that the sentence starts "little is known".  But as currently written, the sentence is syntactically ambiguous (what is the antecedent of "which"?)]

line 422, change "The stylet length of the Boerias spp is shorter than that of" to "The stylets of the Boerias spp are shorter than those of".

line 423, change "lengthier" to "longer"

line 426, change "proven" to "shown"

line 439 change "the predatory Asopiinae subfamily" to "the predatory subfamily Asopiinae".

lines 442, 451.  Don't say that you "generated" the DNA seqs.  Say that you "obtained" them, or "inferred" them.

line 443, change "support" to "supplement"

lines 445-446, change "of which 18 of these" to "of which 18"

line 450, change "several the stink bug" to "several of the stink bug"

lines 467-470, "Genetic diversity was high. . . "  Consider deleting this final sentence of the paragraph.  It is wildly overstated: the fact that genera are highly divergent does not indicate that the morphological groupings are accurate.   The observation is not at all surprising and adds little.

Author Response

Response to Reviewers’ Comments and Suggestions

Please note that the page and line numbers referred to in our responses represent those found in the newly submitted manuscript. Furthermore, specific changes are indicated in italics, where appropriate.

Reviewer 2:

  1. The reviewer suggested that a column should be added to Table 1 for the voucher codes and that the other 28 sequenced specimens be submitted as vouchers.
  • We agree with the reviewer, thus a column for the SANC voucher numbers was added to Table 1. The other 28 sequenced specimens were submitted as vouchers and the voucher codes were added to Table 1.
  • L151: “(PENT00006 -PENT00038)” was changed to “(PENT00006 -PENT00069)”
  1. The reviewer asked how individuals were chosen for sequencing.
  • Two individuals were sequenced for each morphospecies (randomly from different sites), and additional individuals were sequenced for the nine species found in common or in high numbers between regions.
  • L165: “specimen” changed to “morphospecies”
  1. The reviewer indicated that Figure 6 does not accurately represent the non-zero pi values in Table 2 and that only a single DNA sequence represents each species in the phylogenetic analysis.
  • We agree with the reviewer that Figure 6 does not accurately represent the non-zero pi values in Table 2, and we are grateful to the reviewer for picking up the error. The phylogenetic analysis for Figure 6 was redone and now the branches within the species clades accurately represent the non-zero pi values. Each of the species was represented by at least two DNA sequences but the sequence names were listed horizontally. The new phylogeny in Figure 6 displays this better.
  1. The reviewer suggested that a table should be added, as supplementary material, to give precise counts of the morphospecies for each site and season.
  • We agree with the reviewer’s suggestion and have added “Table S1. Number of stink bug morphospecies found at each location across three seasons.” as Supplementary Material.
  • L154: We added “(Table S1)”
  • L224: “3670 stink bug specimens” was changed to “5520 stink bug specimens”. It was noted that this statement only represents the total number of specimens for the 2019-2020 season. The results of the study remain the same as the data for 2017-2018 and 2018-2019 was included in the analyses. We are grateful to the reviewer for suggesting adding Table S1 as it allowed for the detection of this error.
  1. The reviewer offered editing suggestions related to cryptic and incompletely identified species.

5a.) The reviewer suggested changing the sentence in the Simple Summary on lines 21-23 from “A cryptic species of Boerias sp. 1 and Pseudatelus raptorius were also identified.” to “Evidence of cryptic species diversity was also found, within Pseudatelus raptorius and an unidentified Boerias sp. (Boerias sp. 1)”

  • L21-23: “A cryptic species of Boerias 1 and Pseudatelus raptorius were also identified” was replaced with “Evidence of cryptic species diversity was also found, within Pseudatelus raptorius and an unidentified Boerias sp. (Boerias sp. 1).

5b.) The reviewer suggested changing the sentence in the abstract on lines 33-36 from “Boerias sp. 1 Kirkaldy, 1909 and Boerias sp. 2 Kirkaldy, 1909 were the second most abundant species found in KwaZulu-Natal and this is the first report of these species associated with macadamia.” to “Two unidentified species of Boerias Kirkaldy, 1909, here designated as Boerias sp. 1 and Boerias sp. 2, were the second and third most abundant species found in KwaZulu-Natal. No species of Boerias has previously been reported in association with macadamia.”

  • L33-36: “Boerias 1 Kirkaldy, 1909 and Boerias sp. 2 Kirkaldy, 1909 were the second most abundant species found in KwaZulu-Natal and this is the first report of these species associated with macadamia.” was replaced with “Two unidentified species of Boerias Kirkaldy, 1909, here designated as Boerias sp. 1 and Boerias sp. 2, were the second and third most abundant species found in KwaZulu-Natal. No species of Boerias has previously been reported in association with macadamia.

5c.) The reviewer suggested changing the sentence in the abstract on line 36 from “A third cryptic Boerias sp. was also identified on macadamia in this region” to “Evidence of a cryptic third species of Boerias was also found”

  • L36: “A third cryptic Boerias sp. was also identified on macadamia in this region” was replaced with “Evidence of a cryptic third species of Boerias was also found.

6.)  The reviewer suggested that Table 3 be omitted.

  • We agree with the suggestion made by the reviewer and Table 3 was omitted.

7.) The reviewer suggested a number of minor suggestions and copy-edits

  • Line 32: we added “,” after 1878
  • Line 48: “Hawaii” was changed to “USA (Hawaii)
  • Lines 57-58: we removed “,” between the species names and author names. “Linnaeus” was removed after “Corylus sp.”
  • Line 63: “previous” was changed to “previously
  • Lines 68: We omitted “The species identity of some observed species is however unknown”
  • Line 72: We omitted “approaches”
  • Lines 76: We changed “DNA sequence database” to “a DNA sequence database
  • Line 86-87: “the species presence” was changed to “the presence or absence of each species
  • Line 148: “in literature” was changed to “in the literature
  • Line 230: “to species-level” was changed to “to species
  • Line 334: “Individuals of each species group clustered well within their respective branches” was changed to “Individuals of each species clustered within their respective clade”. The error pointed out by the reviewer in point #3 has been rectified and clusters are now represented in Figure 6.
  • Line 410: We omitted “which requires further investigation”
  • Lines 410-411: “The stylet length of the Boerias is shorter than that of” was changed to “The stylets of the Boerias spp. are shorter than those of
  • Line 412: “lengthier” was changed to “longer
  • Line 414: “proven” was changed to “shown
  • Lines 427-428: “the predatory Asopinae subfamily” was changed to “the predatory subfamily Asopinae
  • Line 431: “generated” was changed to “obtained
  • Line 440: “generated” was changed to “obtained
  • Line 432: “support” was changed to “supplement
  • Line 435: We omitted “of these”
  • Line 439: We added “of” after several
  • Line 556: We omitted “Genetic diversity was high (>11.4%) between species in different genera in this study, thus indicating that all of the groups analysed for sequence divergence were in fact separate species and the grouping based on morphology was accurate.”

Reviewer 3 Report

The paper entitled, “Diversity and molecular barcoding of stink bugs (Hemiptera: Pentatomidae) associated with macadamia in South Africa” showcases relevant and useful results on the identification and diversity of different stink bug species found in three major macadamia-growing regions of South Africa.

Below are some notes to better enhance the paper’s readability and chances of getting accepted for publication.

Methods:

The paper discusses the molecular aspect of the study more so than the morphologically-based identification approach. The collection method was not very clearly stated. What spray materials were used? The active ingredient used may have an effect on the species of stink bugs dying and ending up on the plastic sheets. How soon after the spraying did the collection occur? How often was it done per location?

What taxonomic keys were used for the morphological identification?

Results:

The paper talked about the dominance/prevalance of certain species over the other but did not provide the numbers to support these results. The same is true for their seasonal/phenology results. It’d be great to highlight these results using a graph or table.

In its current form, the paper appears to be more descriptive than it is being quantitative. A closer examination of the data sets and a strategic re-presentation of the results may help strengthen the paper.

Author Response

Reviewer 3:

  1. The reviewer asked what spray materials were used for the collection of stink bug specimens and how soon after the spraying did the collection occur? They also asked how often was it done per location?
  • Broad spectrum pyrethroids were applied during farmer scouting for the collection of stink bug specimens. The insecticides are sprayed before dawn and the stink bugs are collected within 3-4 hours after spraying. Farmer scouting is variable between farms and is usually performed to inform their pesticide application decisions.
  • L95-96: We added “a broad-spectrum insecticide spray (i.e., pyrethroid)
  • L96-97: We added “before dawn
  • L97-98: We added “three to four hours after spraying
  1. The reviewer asked what taxonomic keys were used for the morphological identification?
  • Taxonomic keys were only available in the literature for some of the species found in this study. These are referenced in L148. The other species, for which keys are not readily available, were identified based on comparisons of morphological characteristics to known/verified species in the in the National Collection of Insects, Pretoria, Agricultural Research Council - Plant Health and Protection.
  • L148: We added “in taxonomic keys available
  • L149-154: We added “The other species, for which keys are not readily available, were identified based on comparisons of morphological characteristics to verified species in the in the National Collection of Insects, Pretoria, Agricultural Research Council - Plant Health and Protection. Examples of these morphological characteristics include the body size, body colour, antennae, stylets, connexivum, wing membrane, scutellum, pronotum, anterior and posterior margins, and legs.”
  1. The reviewer asked if the results relating to relative stink bug numbers and their seasonal presence could be represented using a table or graph.
  • We agree with the reviewer and due to the same suggestion made by Reviewer 2 a table representing the relative numbers was added to Supplementary Materials (Table S1).
  • We added “Table S2. Seasonal presence of stink bug species per scout batch per region.” to Supplementary Materials to represent the seasonal presence of stink bug species found in scout batches across all three growing regions.
  • L160: We added “(Table S2)”.
  • L459-460: We added “Table S2: Seasonal presence of stink bug species per scout batch per region;

  1. The reviewer suggested that in its current form, the paper appears to be more descriptive than quantitative.

We agree with the reviewer that the paper is more descriptive. We aimed to look at the total species presence between growing regions and seasons as well as establishing a COI database of these species. We were limited in the quantitative and statistical inferences that could be made as we relied on farmer scouting for sampling to obtain stink bug specimens. Scouting regimes occur variably between farms, where some scout more than others, and is dependent on different management decisions between farms. We feel that it would be incorrect or misleading to make too many quantitative inferences that could not be supported statistically due to the variability in scout batches between farms.

  1. L48: The reviewer suggested changing “Hawaii” to “US (particularly Hawaii)”
  • We agree with the reviewer and this error has been corrected based on the suggestion made by Reviewer 2.
  1. The reviewer indicated that the sentence on lines 76-77 is a bit unclear.
  • We appreciate the reviewer pointing out the error which has been corrected as suggested by Reviewer 2.
  1. The reviewer suggested presenting the results mentioned in lines 154-155 in graphical format.
  • We agree that this is a good suggestion and a supplementary table “Table S1: Number of stink bug morphospecies found at each location across three seasons.” was added as also suggested by Reviewer 2.
  1. The reviewer asked if high resolution photos will be included in the final version
  • High resolution photos are available to be submitted individually to ensure clear images are included in the final version.
  1. The reviewer suggested that Table 1 be included as Supplementary Material.
  • We agree with the suggestion made by the reviewer and Table 1 has been included in Supplementary Materials as Table S3.
  • L319: “Table 1” was changed to “Table S3
  • L460-461: We added “Table S3: Collection details of specimens sequenced to determine species presence and composition of stink bugs in macadamia orchards in South Africa.
  1. L257-258: The reviewer asked what the percentage of distincta was compared to the other species listed. The reviewer asked if the relative numbers can be shown to support the results.
  • L261-262: “all growing regions” was changed to “Limpopo (88%), KwaZulu-Natal (76%) and Mpumalanga (62%)
  • The relative numbers can be found in the newly added Table S1.
  • L262: We added “(Table S1)
  1. The reviewer asked what the relative numbers are for Nezara viridula mentioned in lines 306-307.
  • The relative numbers can be found in the newly added Table S1.
  • Lines 307: We added “(Table S1)
  1. The reviewer indicated that the line numbers are overlapping with the text in Table 1.
  • The line numbers were removed and no longer overlap with the text of Table 1